# SPROUT: SELF-PROGRESSING ROBUST TRAINING

## ABSTRACT

Enhancing model robustness under new and even adversarial environments is a crucial milestone toward building trustworthy and reliable machine learning systems. Current robust training methods such as adversarial training explicitly specify an "attack" (e.g., $\ell_\infty$-norm bounded perturbation) to generate adversarial examples during model training in order to improve adversarial robustness. In this paper, we take a different perspective and propose a new framework SPROUT, self-progressing robust training. During model training, SPROUT progressively adjusts training label distribution via our proposed parametrized label smoothing technique, making training free of attack generation and more scalable. We also motivate SPROUT using a general formulation based on vicinity risk minimization, which includes many robust training methods as special cases. Compared with state-of-the-art adversarial training methods (PGD-$\ell_\infty$ and TRADES) under $\ell_\infty$-norm bounded attacks and various invariance tests, SPROUT consistently attains superior performance and is more scalable to large neural networks. Our results shed new light on scalable, effective and attack-independent robust training methods.

## 1 INTRODUCTION

While deep neural networks (DNNs) have achieved unprecedented performance on a variety of datasets and across domains, developing better training algorithms that are capable of strengthening model robustness is the next crucial milestone toward trustworthy and reliable machine learning systems. In recent years, DNNs trained by standard algorithms (i.e., the natural models) are shown to be vulnerable to adversarial input perturbations (Biggio et al., 2013; Szegedy et al., 2014). Adversarial examples crafted by designed input perturbations can easily cause erroneous decision making of natural models (Goodfellow et al., 2015) and thus intensify the demand for robust training methods.

State-of-the-art robust training algorithms are primarily based on the methodology of adversarial training (Goodfellow et al., 2015; Madry et al., 2018), which calls specific attacking algorithms to generate adversarial examples during model training in order to learn robust models. Albeit effective, adversarial training based methods have the following limitations: (i) *poor scalability* – the process of generating adversarial examples incurs considerable computation overhead. For instance, our own experiments show that, with the same computation resources, standard adversarial training (with 7 attack iterations per sample in every minibatch) of Wide ResNet on CIFAR-10 consumes 10 times more clock time per training epoch when compared with standard training; (ii) *attack specificity* – adversarially trained models are usually most effective against the same attack they trained on, and the robustness may not generalize well to other types of attacks (Tramèr & Boneh, 2019; Kang et al., 2019); (iii) *preference toward wider network* – adversarial training are more effective when the networks have sufficient capacity (e.g., having more neurons in network layers) (Madry et al., 2018).

To address the aforementioned limitations, in this paper we propose a new robust training method named SPROUT, which is short for self-progressing robust training. We motivate SPROUT by introducing a general framework that formulates robust training objectives via vicinity risk minimization (VRM), which includes many robust training methods as special cases. It is worth noting that the robust training methodology of SPROUT is fundamentally different from adversarial training, as SPROUT features self-adjusted label distribution during training instead of attack generation. In addition to our proposed parametrized label smoothing technique for progressive adjustification of training label distribution, SPROUT also adopts Gaussian augmentation and Mixup (Zhang et al., 2018) to further enhance robustness. In contrast to adversarial training, SPROUT spares the need for attack generation and thus makes its training scalable by a significant factor, while attaining better or

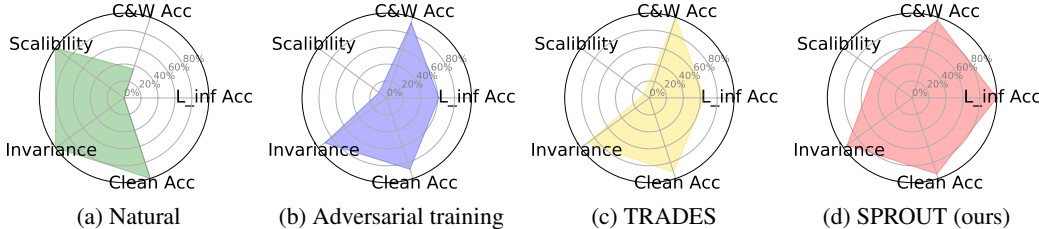

Figure 1: Multi-dimensional performance comparison of four training methods using VGG-16 network and CIFAR-10 dataset. All dimensions are separately normalized by the best-performance method. The average score of each method is 0.6718 for natural (standard training), 0.6900 for PGD-$\ell_\infty$ based adversarial training (Madry et al., 2018), 0.7107 for PGD-$\ell_\infty$ based TRADES (Zhang et al., 2019), and 0.8798 for SPROUT (ours). The exact numbers are reported in Table 6 in Appendix.

comparable robustness performance on a variety of experiments. We also show that SPROUT can find robust models from either randomly initialized models or pretrained models, and its robustness performance is less sensitive to network width.

## 1.1 CONTRIBUTIONS

**Multi-dimensional performance enhancement.** To illustrate the advantage of SPROUT over adversarial training and its variations, Figure 1 compares the model performance of different training methods with the following five dimensions summarized from our experimental results: (i) *Clean Acc* – standard test accuracy, (ii) *L_inf Acc* – accuracy under $\ell_\infty$-norm projected gradient descent (PGD) attack (Madry et al., 2018), (iii) *C&W Acc* – accuracy under $\ell_2$-norm Carlini-Wagner (C&W) attack, (iv) *scalability* – per epoch clock run-time, and (v) *invariance* – invariant transformation tests including rotation, brightness, contrast and gray images. Comparing to PGD-$\ell_\infty$ based adversarial training (Madry et al., 2018) and TRADES (Zhang et al., 2019), SPROUT attains at least 20% better L_inf Acc, 2% better Clean Acc, 5× faster run-time (scalability), 2% better invariance, while maintaining C&W Acc, suggesting a new robust training paradigm that is scalable and comprehensive.

We further summarize the main contributions of this paper as follows:
• We propose SPROUT, a self-progressing robust training method composed of three modules that are efficient and free of attack generation: parametrized label smoothing, Gaussian augmentation and Mixup (Section 3). They altogether attain the state-of-the-art robustness performance and are scalable to large-scale networks. In Section 4.6 we will show that these modules are complimentary to enhancing robustness. We also perform an ablation study to demonstrate that our proposed parametrized label smoothing technique contributes to the major gain in boosting robustness.
• To provide technical explanations for SPROUT, in Section 2 we motivate its training methodology based on the framework of vicinity risk minimization (VRM). We show that many robust training methods, including attack-specific and attack-independent approaches, can be characterized as a specific form of VRM. The superior empirical results of SPROUT provide new insights on developing efficient robust training methods and theoretical justification based on VRM.
• We evaluate the multi-dimensional performance of different training methods on (wide) ResNet and VGG networks using CIFAR-10 and ImageNet datasets. Notably, although SPROUT is attack-independent during training, we find that SPROUT significantly outperforms two major adversarial training methods, PGD-$\ell_\infty$ adversarial training (Madry et al., 2018) and TRADES (Zhang et al., 2019), against the same type of attacks they used during training (Section 4.2). Moreover, SPROUT is more scalable and runs at least 5× faster than adversarial training methods (Section 4.5). It also attains higher clean accuracy and generalizes better to various invariance tests (Section 4.4) and is less sensitive to network width (Section 4.6).

## 1.2 RELATED WORK

**Attack-specific robust training.** The seminal work of adversarial training with a first-order attack algorithm for generating adversarial examples (Madry et al., 2018) has greatly improved adversarial robustness under the same threat model (e.g., $\ell_\infty$-norm bounded perturbations) as the attack algorithm. It has since inspired many advanced adversarial training algorithms with improved robustness. For instance, TRADES (Zhang et al., 2019) is designed to minimize a theoretically-driven upper bound

on prediction error in adversarial examples, which led to the first-ranked defense in the NeurIPS 2018 Adversarial Vision Challenge. Bilateral adversarial training (Wang, 2019) finds robust models by adversarially perturbing the data samples as well as the associated data labels using attack algorithms. A feature-scattering based adversarial training method is proposed in (Zhang & Wang, 2019). Different from attack-specific robust training methods, our proposed SPROUT is free of attack generation, yet it can outperform attack-specific methods. Another line of recent works uses an adversarially trained model along with additional unlabeled data (Carmon et al., 2019; Stanforth et al., 2019) or self-supervised learning with adversarial examples (Hendrycks et al., 2019) to improve robustness, which in principle can also be used in SPROUT but is beyond the scope of this paper.

**Attack-independent robust training.** Here we discuss related works on Gaussian data augmentation, Mixup and label smoothing. Gaussian data augmentation during training is a commonly used baseline method to improve model robustness (Zantedeschi et al., 2017). It is revisited in (Cohen et al., 2019) as a scalable and certifiable defense method called random smoothing. Mixup (Zhang et al., 2018) and its variants (Verma et al., 2018; Thulasidasan et al., 2019) are a recently proposed approach to improve model robustness and generalization by training a model on convex combinations of data sample pairs and their labels. Label smoothing was originally proposed in (Szegedy et al., 2016) as a regularizer to stabilize model training. The main idea is to replace one-hot encoded labels by assigning non-zero (e.g., uniform) weights to every label other than the original training label. Although label smoothing is also shown to benefit model robustness (Shafahi et al., 2018; Goibert & Dohmatob, 2019), its robustness gain is relatively marginal when compared to adversarial training. In contrast to current static (i.e., pre-defined) label smoothing function, in SPROUT we propose a novel parametrized label smoothing scheme, which enables adaptive sampling of training labels from a parameterized distribution on the label simplex. The parameters of the label distribution are progressively adjusted according to the updates of model weights.

## 2 GENERAL FRAMEWORK FOR FORMULATING ROBUST TRAINING

In supervised learning, the task is essentially learning a $K$-class classification function $f \in \mathcal{F}$ that has a desirable mapping between a data sample $\boldsymbol{x} \in \mathcal{X}$ and the corresponding label $\boldsymbol{y} \in \mathcal{Y}$. Consider a loss function $L$ that penalizes the difference between the prediction $f(\boldsymbol{x})$ and the true label $\boldsymbol{y}$ from an unknown data distribution $P$, $(\boldsymbol{x}, \boldsymbol{y}) \sim P$. The population risk can be expressed as

$$R(f) = \int L(f(\boldsymbol{x}), \boldsymbol{y}) P(\boldsymbol{x}, \boldsymbol{y}) d\boldsymbol{x} d\boldsymbol{y} \tag{1}$$

However, as the distribution $P$ is unknown, in practice machine learning uses empirical risk minimization (ERM) with the empirical data distribution of $n$ training data $\{x_i, y_i\}_{i=1}^n$

$$P_\delta(\boldsymbol{x}, \boldsymbol{y}) = \frac{1}{n} \sum_{i=1}^n \delta(\boldsymbol{x} = \boldsymbol{x}_i, \boldsymbol{y} = \boldsymbol{y}_i) \tag{2}$$

to approximate $P(\boldsymbol{x}, \boldsymbol{y})$, where $\delta$ is a Dirac mass. However, a more principled approach is to use Vicinity Risk Minimization (VRM) (Chapelle et al., 2001), which is

$$P_\nu(\boldsymbol{x}, \boldsymbol{y}) = \frac{1}{n} \sum_{i=1}^n \nu(\tilde{\boldsymbol{x}}, \tilde{\boldsymbol{y}} | \boldsymbol{x}_i, \boldsymbol{y}_i) \tag{3}$$

where $\nu$ is a vicinity distribution that measures the probability of finding the virtual sample-label pair $(\tilde{\boldsymbol{x}}, \tilde{\boldsymbol{y}})$ in the vicinity of the training sample-label pair $(\boldsymbol{x}_i, \boldsymbol{y}_i)$. Therefore, ERM can be viewed as a special case of VRM when $\nu = \delta$. VRM has also been used to motivate Mixup training (Zhang et al., 2018). Based on VRM, we propose a general framework that encompasses the objectives of many robust training methods as the following generalized cross entropy loss:

$$H(\tilde{\boldsymbol{x}}, \tilde{\boldsymbol{y}}, f) = - \sum_{k=1}^K [\log g(f(\tilde{\boldsymbol{x}})_k)] h(\tilde{y}_k) \tag{4}$$

where $f(\tilde{\boldsymbol{x}})_k$ is the model's $k$-th class probability output of the input $\tilde{\boldsymbol{x}}$, $g(\cdot) : \mathbb{R} \to \mathbb{R}$ is a mapping adjusting the output probability, and $h(\cdot) : \mathbb{R} \to \mathbb{R}$ is a mapping adjusting the training label distribution. When $\tilde{\boldsymbol{x}} = \boldsymbol{x}$, $\tilde{\boldsymbol{y}} = \boldsymbol{y}$ and $g = h = \mathcal{I}$, where $\mathcal{I}$ denotes the identity mapping function, the loss in (4) degenerates to the conventional cross entropy loss used in ERM.

Table 1: Summary of several robust training methods using VRM formulation in (4). $\text{PGD}_\epsilon(\cdot)$ means (multi-step) PGD attack with perturbation budget $\epsilon$ and Dirichlet($\mathbf{b}$) is the Dirichlet distribution parameterized by $\mathbf{b}$. GA stands for Gaussian Augmentation and LS stands for Label Smoothing.

| Methods | $g(\cdot)$ | $h(\cdot)$ | $\tilde{x}$ | $\tilde{y}$ | attack-specific |
|---|---|---|---|---|---|
| Natural | $\mathcal{I}$ | $\mathcal{I}$ | $x$ | $y$ | × |
| GA (Zantedeschi et al., 2017) | $\mathcal{I}$ | $\mathcal{I}$ | $\mathcal{N}(x, \Delta^2)$ | $y$ | × |
| LS (Szegedy et al., 2016) | $\mathcal{I}$ | $(1-\alpha)y + \alpha u$ | $x$ | $y$ | × |
| Adversarial training (Madry et al., 2018) | $\mathcal{I}$ | $\mathcal{I}$ | $\text{PGD}_\epsilon(x)$ | $y$ | ✓ |
| TRADES (Zhang et al., 2019) | $\mathcal{I}$ | $(1-\alpha)y + \alpha f(\tilde{x})$ | $\text{PGD}_\epsilon(x)$ | $y$ | ✓ |
| Stable training (Zheng et al., 2016) | $f(x) \circ f(\tilde{x})$ | $\mathcal{I}$ | $\mathcal{N}(x, \Delta^2)$ | $y$ | × |
| Mixup (Zhang et al., 2018) | $\mathcal{I}$ | $\mathcal{I}$ | $(1-\lambda)x_i + \lambda x_j$ | $(1-\lambda)y_i + \lambda y_j$ | × |
| LS+GA (Shafahi et al., 2018) | $\mathcal{I}$ | $(1-\alpha)y + \alpha u$ | $\mathcal{N}(x, \Delta^2)$ | $y$ | × |
| Bilateral Adv Training (Wang, 2019) | $\mathcal{I}$ | $\mathcal{I}$ | $\text{PGD}_\epsilon(x)$ (one or two step) | $(1-\alpha)y_i + \alpha\text{PGD}_{\epsilon'}(y)$ | ✓ |
| SPROUT (ours) | $\mathcal{I}$ | Dirichlet$((1-\alpha)y + \alpha\beta)$ | $(1-\lambda)\mathcal{N}(x_i, \Delta^2) + \lambda\mathcal{N}(x_j, \Delta^2)$ | $(1-\lambda)y_i + \lambda y_j$ | × |

Based on the general VRM loss formulation in (4), in Table 1 we summarize a large body of robust training methods in terms of different expressions of $g(\cdot)$, $h(\cdot)$ and $(\tilde{x}, \tilde{y})$. For example, the vanilla adversarial training in (Madry et al., 2018) aims to minimize the loss of adversarial examples generated by the (multi-step) PGD attack with perturbation budget $\epsilon$, denoted by $\text{PGD}_\epsilon(\cdot)$. Its training objective can be rewritten as $\tilde{x} = \text{PGD}_\epsilon(x)$, $\tilde{y} = y$ and $g = h = \mathcal{I}$. In addition to adversarial training only on perturbed samples of $x$, Wang (2019) designs adversarial label perturbation where it uses $\tilde{x} = \text{PGD}_\epsilon(x)$, $\tilde{y} = (1-\alpha)y + \alpha\text{PGD}_\epsilon(y)$, and $\alpha \in [0, 1]$ is a mixing parameter. TRADES (Zhang et al., 2019) improves adversarial training with an additional regularization on the clean examples, which is equivalent to replacing the label mapping function $h(\cdot)$ from identity to $(1-\alpha)y + \alpha f(\tilde{x})$. Label smoothing (LS) alone is equivalent to the setup that $g = \mathcal{I}$, $\tilde{x} = x$, $\tilde{y} = y$ and $h(\cdot) = (1-\alpha)y + \alpha u$, where $u$ is often set as a uniform vector with value $1/K$ for a $K$-class supervised learning task. Joint training with Gaussian augmentation (GA) and label smoothing (LS) as studied in (Shafahi et al., 2018) is equivalent to the case when $\tilde{x} = \mathcal{N}(x, \Delta^2)$, $\tilde{y} = y$, $g = \mathcal{I}$ and $h(y) = (1-\alpha)y + \alpha/K$. We defer the connection between SPROUT and VRM to the next section.

## 3 SPROUT: Scalable Robust and Generalizable Training

In this section, we formally introduce SPROUT, a novel robust training method that automatically finds a better vicinal risk function during model training in a self-progressing manner.

### 3.1 Self-Progressing Training via Parametrized Label Smoothing

To stabilize training and improve model generalization, Szegedy et al. (2016) introduces label smoothing that converts "one-hot" label vectors into "one-warm" vectors representing low-confidence classification, in order to prevent a model from making over-confident predictions. Specifically, the one-hot encoded label $y$ is smoothed using

$$\tilde{y} = (1-\alpha)y + \alpha u \tag{5}$$

where $\alpha \in [0, 1]$ is the smoothing parameter. A common choice is the uniform distribution $u = \frac{1}{K}$, where $K$ is the number of classes. Later works (Wang, 2019; Goibert & Dohmatob, 2019) use an attack-driven label smoothing function $u$ to further improve adversarial robustness. However, both uniform and attack-driven label smoothing disregard the inherent correlation between labels. To address the label correlation, we propose to use the Dirichlet distribution parametrized by $\beta \in \mathbb{R}_+^K$ for label smoothing. Our SPROUT learns to update $\beta$ to find a training label distribution that is most uncertain to current model weights $\theta$, by solving

$$\max_\beta L(\tilde{x}, \tilde{y}, \beta; \theta) \tag{6}$$

where $\tilde{y} = \text{Dirichlet}((1-\alpha)y + \alpha\beta)$. Notably, instead of using a pre-defined or attack-driven function for $u$ in label smoothing, our Dirichlet label smoothing approach automatically finds a label simplex by optimizing $\beta$. Dirichlet distribution indeed takes label correlation into consideration as its generated label $z = [z_1, \ldots, z_K]$ has the statistical properties

$$\mathbb{E}[z_s] = \frac{\beta_s}{\beta_0}, \; \text{Cov}[z_s, z_t] = \frac{-\beta_s\beta_t}{\beta_0^2(\beta_0+1)}, \; \sum_{s=1}^K z_s = 1 \tag{7}$$

where $\beta_0 = \sum_{k=1}^K \beta_k$ and $s, t \in \{1, \ldots, K\}$, $s \neq t$. Moreover, one-hot label and uniform label smoothing are special cases of our Dirichlet label smoothing when $\beta = y$ and $\beta = u$, respectively.

Our Dirichlet label smoothing co-trains with the update in model weights $\theta$ during training (see Algorithm 1). The advantage of our proposed self-progressing Dirichlet label smoothing over uniform label smoothing will be justified in our ablation study (see Figure 5 in Section 4.6). In addition, we illustrate the label correlation learned from our Dirichlet label smoothing in Appendix A.2.

## 3.2 INCORPORATING GAUSSIAN DATA AUGMENTATION AND MIXUP

**Gaussian augmentation.** Adding Gaussian noise to data samples during training (i.e., Gaussian augmentation) is a common pracitice to improve model robustness. Its corresponding vicinal function is the Gaussian vicinity function $\nu(\tilde{\boldsymbol{x}}_i, \tilde{\boldsymbol{y}}_i | \boldsymbol{x}_i, \boldsymbol{y}_i) = \mathcal{N}(\boldsymbol{x}_i, \Delta^2)\delta(\tilde{\boldsymbol{y}}_i = \boldsymbol{y}_i)$, where $\Delta^2$ is the variance of a standard normal random vector. However, the gain of Gaussian augmentation in robustness is marginal when compared with adversarial training (see our ablation study in Section 4.6). Shafahi et al. (2018) finds that combining uniform or attack-driven label smoothing with Gaussian smoothing can further improve adversarial robustness. Therefore, we propose to incorporate Gaussian augmentaion with Dirichlet label smoothing. The joint vicinity function becomes

$$\nu(\tilde{\boldsymbol{x}}_i, \tilde{\boldsymbol{y}}_i | \boldsymbol{x}_i, \boldsymbol{y}_i, \boldsymbol{\beta}) = \mathcal{N}(\boldsymbol{x}_i, \Delta^2)\delta(\tilde{\boldsymbol{y}}_i = \text{Dirichlet}((1-\alpha)\boldsymbol{y}_i + \alpha\boldsymbol{\beta})) \tag{8}$$

Training with equation (8) means drawing labels from the parameterized Dirichlet distribution for the original data sample $\boldsymbol{x}_i$ and its neighborhood characterized by Gaussian augmentation.

**Mixup.** To further improve model generalization, SPROUT also integrates Mixup (Zhang et al., 2018) that performs convex combination on pairs of training data samples (in a minibatch) and their labels during training. The vicinity function of Mixup is $\nu(\tilde{\boldsymbol{x}}, \tilde{\boldsymbol{y}} | \boldsymbol{x}_i, \boldsymbol{y}_i) = \delta(\tilde{\boldsymbol{x}} = (1-\lambda)\boldsymbol{x}_i + \lambda\boldsymbol{x}_j, \tilde{\boldsymbol{y}} = (1-\lambda)\boldsymbol{y}_i + \lambda\boldsymbol{y}_j)$, where $\lambda \sim \text{Beta}(a, a)$ is the mixing parameter drawn from the Beta distribution and $a > 0$ is the shape parameter. The Mixup vicinity function can be generalized to multiple data sample pairs. Unlike Gaussian augmentation which is irrespective of the label (i.e., only adding noise to $\boldsymbol{x}_i$), Mixup aims to augment data samples on the line segments of training data pairs and assign them convexly combined labels during training.

**Vicinity function of SPROUT.** With the aforementioned techniques integrated in SPROUT, the overall vicinity function of SPROUT can be summarized as

$$\nu(\tilde{\boldsymbol{x}}, \tilde{\boldsymbol{y}} | \boldsymbol{x}_i, \boldsymbol{y}_i, \boldsymbol{\beta}) \tag{9}$$
$$= \delta(\tilde{\boldsymbol{x}} = \lambda\mathcal{N}(\boldsymbol{x}_i, \Delta^2) + (1-\lambda)\mathcal{N}(\boldsymbol{x}_j, \Delta^2), \tilde{\boldsymbol{y}} = \text{Dirichlet}((1-\alpha)((1-\lambda)\boldsymbol{y}_i + \lambda\boldsymbol{y}_j) + \alpha\boldsymbol{\beta})$$

In Section 4.6, we will show that Dirichlet label smoothing, Gaussian augmentation and Mixup are complimentary to enhancing robustness by showing their diversity in input gradients.

## 3.3 SPROUT ALGORITHM

Using the VRM framework, the training objective of SPROUT is

$$\min_{\theta} \max_{\boldsymbol{\beta}} \sum_{i=1}^{n} L(\nu(\tilde{\boldsymbol{x}}_i, \tilde{\boldsymbol{y}}_i | \boldsymbol{x}_i, \boldsymbol{y}_i, \boldsymbol{\beta}); \theta), \tag{10}$$

where $\theta$ denotes the model weights, $n$ is the number of training data, $L$ is the generalized cross entropy loss defined in (4) and $\nu(\tilde{\boldsymbol{x}}, \tilde{\boldsymbol{y}} | \boldsymbol{x}_i, \boldsymbol{y}_i, \boldsymbol{\beta})$ is the vicinity function defined in (9). Our SPROUT algorithm co-trains $\theta$ and $\boldsymbol{\beta}$ via stochastic gradient descent/ascent to solve the outer minimization problem on $\theta$ and the inner maximization problem on $\boldsymbol{\beta}$. In particular, for calculating the gradient $g_{\boldsymbol{\beta}}$ of the parameter $\boldsymbol{\beta}$, we use the Pytorch implementation based on (Figurnov et al., 2018). SPROUT can either train a model from scratch with randomly initialized $\theta$ or strengthen a pre-trained model. As will be evident in Section 4.2, when evaluated against PGD-$\ell_{\infty}$ attack with different $\epsilon$ perturbation constraints, we find that training from either randomly initialized or pre-trained natural models using SPROUT can yield substantially robust models that are resilient to large $\epsilon$ values. The training steps of SPROUT are summarized in Algorithm 1.

We also note that our min-max training methodology is different from the min-max formulation in adversarial training (Madry et al., 2018), which is $\min_{\theta} \sum_{i=1}^{n} \max_{\boldsymbol{\delta}_i : \|\boldsymbol{\delta}_i\|_p \leq \epsilon} L(\boldsymbol{x}_i + \boldsymbol{\delta}_i, \boldsymbol{y}_i; \theta)$, where $\|\boldsymbol{\delta}_i\|_p$ denotes the $\ell_p$ norm of the adversarial perturbation $\boldsymbol{\delta}_i$. While the outer minimization step for optimizing $\theta$ can be identical, the inner maximization of adversarial training requires running multi-step PGD attack to find adversarial perturbations $\{\boldsymbol{\delta}_i\}$ for each data sample in every minibatch (iteration) and epoch, which is attack-specific and time-consuming (see our scalability analysis in

---

**Algorithm 1** SPROUT algorithm

---

**Input:** Training dataset $(X, Y)$, Mixup parameter $\lambda$, Gaussian augmentation variance $\Delta^2$, model learning rate $\gamma_\theta$, Dirichlet label smoothing learning rate $\gamma_\beta$ and parameter $\alpha$, cross entropy loss $L$
Initial model $\theta$: random initialization (train from scratch) or pre-trained model checkpoint
Initial $\beta$: random initialization
**for** epoch=$1, \ldots, N$ **do**
   **for** minibatch $X_B \subset X, Y_B \subset Y$ **do**
      $X_B \leftarrow \mathcal{N}(X_B, \Delta^2)$
      $X_{mix}, Y_{mix} \leftarrow \text{Mixup}(X_B, Y_B, \lambda)$
      $Y_{mix} \leftarrow \text{Dirichlet}(\alpha Y_{mix} + (1 - \alpha)\beta)$
      $g_\theta \leftarrow \nabla_\theta L(X_{mix}, Y_{mix}, \theta)$
      $g_\beta \leftarrow \nabla_\beta L(X_{mix}, Y_{mix}, \theta)$
      $\theta \leftarrow \theta - \gamma_\theta g_\theta$
      $\beta \leftarrow \beta + \gamma_\beta g_\beta$
   **end for**
**end for**
**return** $\theta$

---

Table 5). On the other hand, our inner maximization is upon the Dirichlet parameter $\beta$, which is attack-independent and only requires single-step stochastic gradient ascent with a minibatch to update $\beta$. We have explored multi-step stochastic gradient ascent on $\beta$ and found no significant performance enhancement but increased computation time.

**Advantages of SPROUT.** Comparing to adversarial training, the training of SPROUT is more efficient and scalable, as it only requires one additional back propagation to update $\beta$ in each iteration (see Table 5 for a run-time analysis). As highlighted in Figure 1, SPROUT is also more comprehensive as it automatically improves robustness in multiple dimensions owing to its self-progressing training methodology. Moreover, we find that SPROUT significantly outperforms adversarial training and is more effective as network width increases (see Figure 7), which makes SPROUT a promising approach to support robust training for a much larger set of network architectures.

## 4 PERFORMANCE EVALUATION

### 4.1 EXPERIMENT SETUP

**Dataset and model structure.** We use CIFAR-10 (Krizhevsky et al.) and ImageNet (Deng et al., 2009) for performance evaluation. For CIFAR-10, we use both standard VGG-16 (Simonyan & Zisserman, 2015) and Wide ResNet that is used in both vanilla adversarial training (Madry et al., 2018) and TRADES (Zhang et al., 2019). The Wide ResNet models are pre-trained PGD-$\ell_\infty$ robust models given by adversarial training and TRADES. For VGG-16, we implement adversarial training with the standard hyper-parameters and train TRADES with the official implementation. For ImageNet, we use ResNet-152. All our experiments were implemented in Pytorch-1.2 and conducted using dual Intel E5-2640 v4 CPUs (2.40GHz) with 512 GB memory with a GTX 1080 GPU.

**Implementation details.** As suggested in Mixup (Zhang et al., 2018), we set the Beta distribution parameter $a = 0.2$ when sampling the mixing parameter $\lambda$. For Gaussian augmentation, we set $\Delta = 0.1$, which is within the suggested range in (Zantedeschi et al., 2017). Also, we set the label smoothing parameter $\alpha = 0.01$. A parameter sensitivity analysis on $\lambda$ and $\alpha$ is given in Appendix A.4. Unless specified otherwise, for SPROUT we set the model initialization to be a natural model. An ablation study of model initialization is given in Section 4.6.

### 4.2 ADVERSARIAL ROBUSTNESS UNDER VARIOUS ATTACKS

**White-box attacks.** On CIFAR-10, we compare the model accuracy under different strength of white-box $\ell_\infty$-norm bounded non-targeted PGD attack, which is considered as the strongest first-order adversary (Madry et al., 2018) with an $\ell_\infty$-norm constraint $\epsilon$ (normalized between 0 to 1). All PGD attacks are implemented with random starts and we run PGD attack with 20 and 100 steps in our experiments. The (robust) accuracy under different $\epsilon$ values are shown in Figure 2. When $\epsilon = 0.03$

and under PGD attack with 20 steps, we find SPROUT achieves 62.24% and 66.23% robust accuracy on VGG16 and Wide ResNet respectively, while TRADES and adversarial training are 10-20% worse than SPROUT. We also find that SPROUT is significantly more robust to PGD-$\ell_\infty$ attacks with large $\epsilon$ values. In addition to the substantially improved robustness, the clean accuracy (i.e., when $\epsilon = 0$) of SPROUT is 5-10 % higher than TRADES and adversarial training, and it is only 2-4% lower than natural model, suggesting SPROUT better balances the robustness-accuracy trade-off. Similar trends are observed in robust accuracy under PGD attack with 100 steps. On Wide ResNet we also report the robust accuracy of the "free adversarial training" (Free Adv train) method (Shafahi et al., 2019), which features similar robust accuracy as adversarial training but greatly reduced training time.

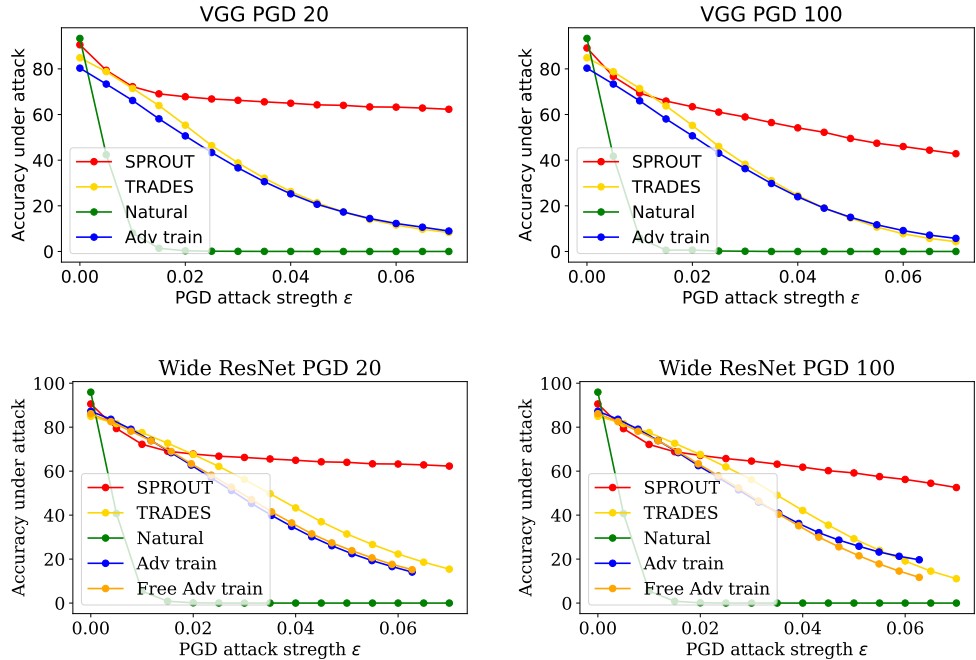

Figure 2: Robust accuracy under PGD-$\ell_\infty$ attack. SPROUT signigicantly outperforms other methods.

In addition to PGD-$\ell_\infty$ attack, we also compare against $\ell_2$-norm based C&W attack (Carlini & Wagner, 2017). We use the default attack setting to do 10 binary search steps with 1000 iterations per step to find successful attacks while minimizing $\ell_2$-norm perturbation. Figure 3 shows that the gain in $\ell_2$ robustness using SPROUT is less apparent than that in $\ell_\infty$ robustness. SPROUT's performance is similar to TRADES but is better than both natural and adversarial training. The results also suggest that the attack-independent and self-progressing training nature of SPROUT can prevent the drawback of failing to provide comprehensive robustness to multiple and simultaneous $\ell_p$-norm adversarial attacks in adversarial training (Tramèr & Boneh, 2019; Kang et al., 2019).

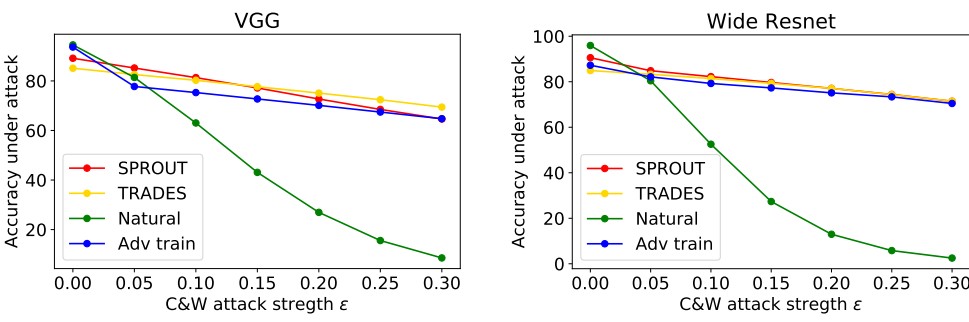

Figure 3: Robust accuracy under C&W-$\ell_2$ attack

**Transfer attack.** We follow the criterion of evaluating transfer attacks as suggested by Athalye et al. (2018) to inspect whether the models trained by SPROUT will cause the issue of obfuscated gradients

and give a false sense of model robustness. We generate 10,000 adversarial examples of CIFAR-10 from natural models with $\epsilon = 0.03$ and evaluate their attack performance on the target model. Table 2 shows SPROUT achieves the best accuracy when compared with adversarial training and TRADES, suggesting the effectiveness of SPROUT in defending both white-box and transfer attacks.

Table 2: Robust accuracy under transfer attack on CIFAR-10

| Method | VGG 16 | Wide ResNet |
|--------|--------|-------------|
| Adv train | 79.13% | 85.84% |
| TRADES | 83.53% | 83.9% |
| SPROUT | 86.28% | 89.1% |

**ImageNet results.** As many ImageNet class labels carry similar semantic meanings, to generate meaningful adversarial examples for robustness evaluation, here we follow the same setup as in (Athalye et al., 2018) that adopts PGD-$\ell_\infty$ attacks with randomly targeted labels. Table 3 compares the robust accuracy of natural and SPROUT models. SPROUT greatly improves the robust accuracy across different $\epsilon$ values. For example, when $\epsilon = 0.01$, SPROUT boosts the robust accuracy of natural model by over 43%. When $\epsilon = 0.015 \approx 4/255$, a considerably large adversarial perturbation on ImageNet, SPROUT still attains about 35% robust accuracy while the natural model merely has about 2% robust accuracy. Moreover, comparing the clean accuracy, SPROUT is about 4% worse than the natural model but is substantially more robust. We omit the comparison to adversarial training methods as we are unaware of any public pre-trained robust ImageNet models prior to the time of our submission, and it is computationally demanding for us to train and fine-tune such large-scale networks with adversarial training. On our machine, training a natural model takes 31,158.7 seconds and training SPROUT takes 59,201.6 seconds. Comparing to the run-time analysis in Section 4.5, SPROUT has a much better scalability than adversarial training and TRADES.

Table 3: Robust accuracy under PGD-$\ell_\infty$ attack on ImageNet and ResNet-152

| Method | Clean Acc | $\epsilon = 0.005$ | $\epsilon = 0.01$ | $\epsilon = 0.015$ | $\epsilon = 0.02$ |
|--------|-----------|--------------------|-------------------|--------------------|--------------------|
| Natural | 78.31% | 37.13% | 9.14% | 2.12% | 0.78% |
| SPROUT | 74.23% | 65.24% | 52.86% | 35.04% | 12.18% |

### 4.3 LOSS LANDSCAPE EXPLORATION

To further verify the superior robustness using SPROUT, we visualize the loss landscape of different training methods in Figure 4. Following the implementation in (Engstrom et al., 2018), we vary the data input along a linear space defined by the sign of the input gradient and a random Rademacher vector, where the x- and y- axes represent the magnitude of the perturbation added in each direction and the z-axis represents the loss. One can observe that the loss surface of SPROUT is smoother. Furthermore, it attains smaller loss variation compared with other robust training methods. The results provide strong evidence for finding more robust models via SPROUT.

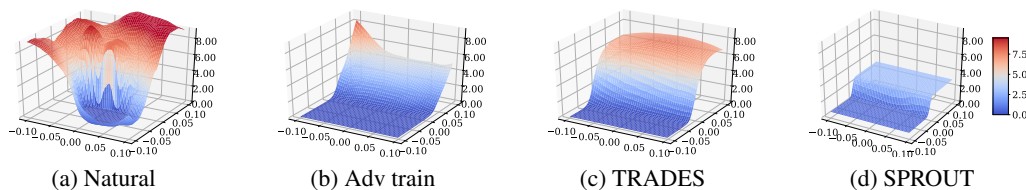

|         (a) Natural         |         (b) Adv train         |         (c) TRADES         |         (d) SPROUT         |

Figure 4: Loss landscape comparison of different training methods

### 4.4 INVARIANCE TEST

In addition to $\ell_p$-norm bounded adversarial attacks, here we also evaluate model robustness against different kinds of input transformations using CIFAR-10 and Wide ResNet. Specifically, we change

rotation (with 10 degree), brightness (increase the brightness factor to 1.5), contrast (increase the contrast factor to 2) and make inputs into grayscale (average all RGB pixel values). The model accuracy under these invariance tests is summarized in Table 4. Results show that SPROUT outperforms adversarial training and TRADES. Interestingly, natural model attains the best accuracy despite the fact that it lacks adversarial robustness, suggesting a potential trade-off between accuracy in these invariance tests and $\ell_p$-norm based adversarial robustness.

Table 4: Accuracy under invariance tests

| Method | Rotation | Brightness | Contrast | Gray |
|---|---|---|---|---|
| Natural | 88.21% | 93.4% | 91.88 % | 91.95% |
| Adv train | 82.66% | 83.64% | 84.99% | 81.08% |
| TRADES | 80.81% | 81.5 % | 83.08% | 79.27% |
| SPROUT | 85.95% | 88.26 % | 86.98% | 81.64% |

Table 5: Training-time (seconds) for 10 epochs

| Methods | CIFAR-10 | |
|---|---|---|
| | VGG 16 | Wide ResNet |
| Natural | 146.7 | 1449.6 |
| Adv train | 1327.1 | 14246.1 |
| TRADES | 1932.5 | 22438.4 |
| SPROUT | 271.7 | 2727.8 |
| Free Adv train(m=8) | 2053.1 | 20652.5 |

## 4.5 SCALABILITY

As illustrated in Section 3.3, SPROUT enjoys great scalability over adversarial training based algorithms because its training requires much less number of back-propagations per iteration, which is a dominating factor that contributes to considerable run-time in adversarial training. Table 5 benchmarks the run-time of different training methods for 10 epochs. On CIFAR-10, the run-time of adverarial training and TRADES is about $5\times$ more than SPROUT. We also report the run-time analysis using the default implementation[1] from the recent work (Free Adv train) in (Shafahi et al., 2019). Its 10-epoch run-time with the replay parameter $m = 8$ is similar to TRADES. However, we note that Free Adv train may require less number of epochs when training to covergence.

## 4.6 ABLATION STUDY

**Dissecting SPROUT.** Here we perform an ablation study using VGG-16 and CIFAR-10 to investigate and factorize the robustness gain in SPROUT's three modules: Dirichlet label smoothing (Dirichlet), Gaussian augmentation (GA) and Mixup. We implement all combinations of these techniques and include uniform label smoothing (LS) (Szegedy et al., 2016) as another baseline. Their accuracies under PGD-$\ell_\infty$ attack are shown in Figure 5. We highlight some important findings as follows.
• Dirichlet outperforms uniform LS by a significant factor, suggesting the importance of our proposed self-progressing label smoothing in improving adversarial robustness.
• Comparing the performance of individual modules alone (GA, Mixup and Dirichlet), our proposed Dirichlet attains the best robust accuracy, suggesting its crucial role in training robust models.
• No other combinations can outperform SPROUT. Moreover, the robust gains from GA, Mixup and Dirichlet appear to be *complimentary*, as SPROUT's accuracy is close to the sum of their individual accuracy. To justify their diversity in robustness, we compute the cosine similarity of their pairwise input gradients and find that they are indeed quite diverse and thus can promote robustness when used together. The details are given in Appendix A.3.

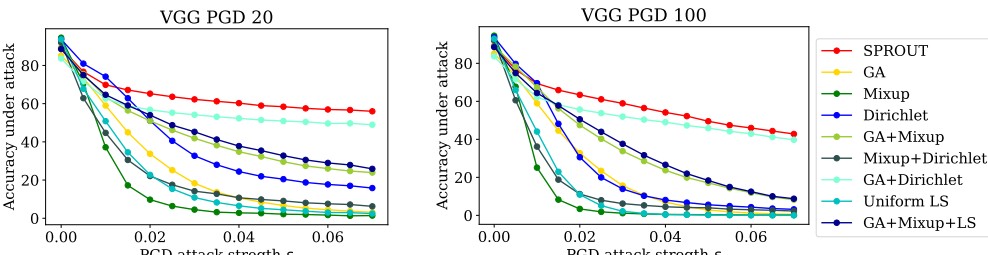

Figure 5: Robust accuracy with different combinations of the modules in SPROUT

**PGD attacks with more iterations.** To ensure the robustness of SPROUT is not an artifact of running insufficient iterations in PGD attack (Athalye et al., 2018), in Figure 6a we show the robust accuracy with the number of PGD-$\ell_\infty$ attack steps varying from 10 to 500 on VGG-16 and CIFAR-10.

---

[1]`https://github.com/mahyarnajibi/FreeAdversarialTraining`

The results show stable performance in SPROUT, TRADES and adversarial training once the number of steps exceeds 100. It is clear that SPROUT indeed outperforms others by a large margin.

**Model weight initialization.** Figure 6b compares the effect of model initialization using CIFAR-10 and VGG-16 under PGD-$\ell_\infty$ attack, where the legend $A + B$ means using Model $A$ as the initialization and training with Method $B$. Interestingly, Natural+SPROUT attains the best robust accuracy when $\epsilon \geq 0.02$. TRADES+SPROUT and Random+SPROUT also exhibit strong robustness since their training objective involves the loss on both clean and adversarial examples. In contrast, Adv train+SPROUT does not have such benefit since adversarial training only aims to minimize adversarial loss. This finding is also unique to SPROUT, as neither Natural+Adv train nor Natural+TRADES can boost robust accuracy. Our results provide novel perspectives and also indicate that SPROUT is indeed a new robust training method that vastly differs from adversarial training based methods.

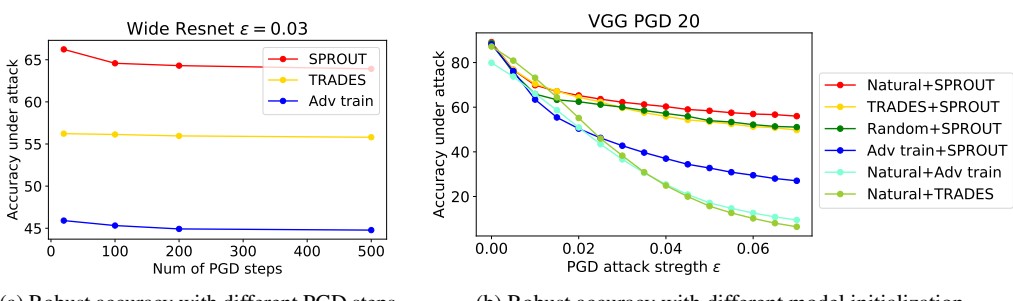

(a) Robust accuracy with different PGD steps      (b) Robust accuracy with different model initialization

Figure 6: Stability in PGD-$\ell_\infty$ attack and the effect of model initialization.

**Effect on network width.** It was shown in (Madry et al., 2018) that adversarial training (Adv train) will take effect when a network has sufficient capacity, which can be achieved by increasing network width. Figure 7 compares the robust accuracy of SPROUT and Adv train with varying network width on Wide ResNet and CIFAR-10. When the network has width = 1 (i.e. a standard ResNet-152 network (He et al., 2016)), the robust accuracy of SPROUT and Adv train are both relatively low (less than 47%). However, as the width increases, SPROUT soon attains significantly better robust accuracy than Adv train by a large margin (roughly 15%). Since SPROUT is more effective in boosting robust accuracy as network width varies, the results also suggest that SPROUT can better support robust training for a broader range of network structures.

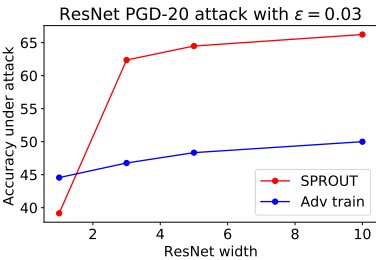

Figure 7: Effect of network width against PGD-$\ell_\infty$ attack on CIFAR-10 and ResNet.

## 5 CONCLUSION

This paper introduced SPROUT, a self-progressing robust training method motivated by vicinity risk minimization. When compared with state-of-the-art adversarial training based methods, our extensive experiments showed that the proposed self-progressing Dirichlet label smoothing technique in SPROUT plays a crucial role in finding substantially more robust models against $\ell_\infty$-norm bounded PGD attacks and simultaneously makes the corresponding model more generalizable to various invariance tests. We also find that SPROUT can strengthen a wider range of network structures as it is less sensitive to network width changes. Moreover, SPOURT's self-adjusted learning methodology not only makes its training free of attack generation but also becomes scalable solutions to large networks. Our results shed new insights on devising comprehensive and robust training methods.

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

# A APPENDIX

## A.1 EXACT PERFORMANCE METRICS FOR FIGURE 1

Table 6: Performance comparison between different training methods on VGG-16 and CIFAR-10

| Method | Clean Acc | $\ell_\infty$ Acc ($\epsilon = 0.03$) | C&W Acc | Invariance (Contrast) | Scalibility (10 epochs) |
|--------|-----------|----------------------|---------|----------------------|-------------------------|
| Natural | 95.93% | 0% | 26.95% | 91.88% | 146.7 (secs) |
| Adv train | 84.92% | 36.29% | 70.13% | 84.99% | 1327.1 (secs) |
| TRADES | 88.6% | 38.29% | 75.08% | 83.08% | 1932.5 (secs) |
| SPROUT | 90.56% | 58.93% | 72.7% | 86.98% | 271.7 (secs) |

## A.2 LEARNED LABEL CORRELATION FROM SPROUT

Based on the statistical properties of Dirichlet distribution in (7), we use the final $\boldsymbol{\beta}$ parameter learned from Algorithm 1 with CIFAR-10 and VGG-16 to display the matrix of its pair-wise product $\beta_s \cdot \beta_t$ in Figure 8. The value in each entry is proportional to the absolute value of the label covariance in (7). We observe some clustering effect of class labels in CIFAR-10, such as relatively high values among the group of {airplane, auto, ship, truck} and relatively low values among the group of {bird, cat, deer, dog}. Moreover, since the $\boldsymbol{\beta}$ parameter is progressively adjusted and co-trained during model training, and the final $\boldsymbol{\beta}$ parameter is clearly not uniformly distributed, the results also validate the importance of using parametrized label smoothing to learn to improve robustness.

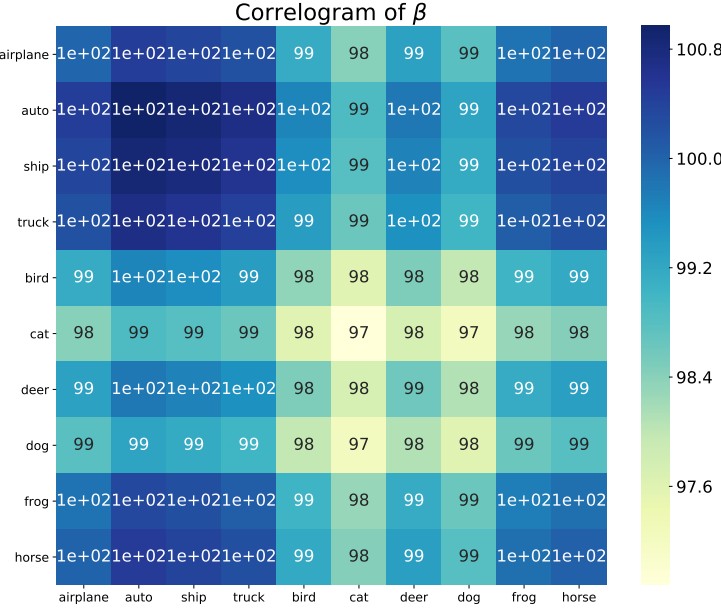

Figure 8: Matrix plot of the product $\beta_s \cdot \beta_t$ of the learned $\boldsymbol{\beta}$ parameter on CIFAR-10 and VGG-16.

## A.3 DIVERSITY ANALYSIS

In order to show the three modules (Dirichlet LS, GA and Mixup) in SPROUT lead to robustness gains that are complimentary to each other, we perform a diversity analysis motivated by (Kariyappa & Qureshi, 2019) to measure the similarity of their pair-wise input gradients and report the average cosine similarity in Table 7 over 10,000 data samples using CIFAR-10 and VGG-16. We find that the pair-wise similarity between modules is indeed quite small ($< 0.103$). The Mixup-GA similarity is the smallest among all pairs since the former performs both label and data augmentation based on convex combinations of training data, whereas the latter only considers random data augmentation. The Dirichlet_LS-GA similarity is the second smallest (and it is close to the Mixup-GA similarity) since the former progressively adjusts the training label $\tilde{y}$ while the latter only randomly adjusts the training sample $\tilde{x}$. The Dirichlet_LS-Mixup similarity is relatively high because Mixup depends on the training samples and their labels while Dirichlet LS also depend on them and the model weights. The results show that their input gradients are diverse as they point to vastly different directions. Therefore, SPROUT enjoys complimentary robustness gain and can promote robustness when combining these techniques together.

Table 7: Average pair-wise cosine similarity of the three modules in SPROUT

|  | Dirichilet LS | Mixup | GA |
|---|---|---|---|
| Dirichilet LS | NA | 0.1023 | 0.0163 |
| Mixup | 0.1023 | NA | 0.0111 |
| GA | 0.0163 | 0.0111 | NA |

## A.4 PARAMETER SENSITIVITY ANALYSIS

We perform an sensitivity analysis of the mixing parameter $\lambda \sim \text{Beta}(a,a)$ and the smoothing parameter $\alpha$ of SPROUT in Figure 9. When fixing $a$, we find that setting $\alpha$ too large may affect robust accuracy, as the resulting training label distribution could be too uncertain to train a robust model. Similarly, when fixing $\alpha$, setting $a$ too large may also affect robust accuracy.

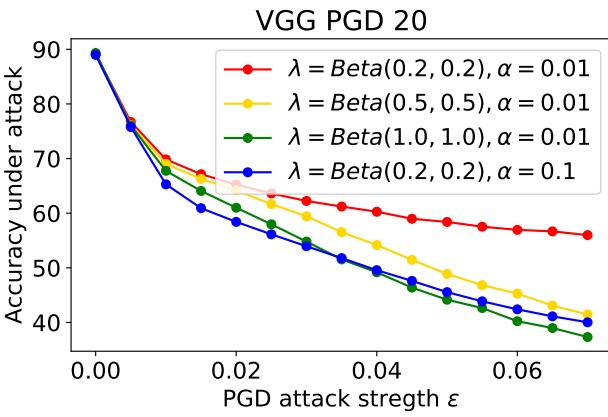

Figure 9: Sensitivity of hyperparameters $\lambda$ and $\alpha$ in SPROUT under PGD-$\ell_\infty$ attack

## A.5 PERFORMANCE ON CW-$\ell_\infty$ ATTACK

To further test the robustness on $\ell_\infty$ constraint, we replace the cross entropy loss with CW-$\ell_\infty$ loss (Carlini & Wagner, 2017) in PGD attack. Similar to the PGD-$\ell_\infty$ attack results in Figure 2, Figure 10 shows that although SPROUT has slightly worse accuracy under small $\epsilon$ values, it attains much higher robust accuracy when $\epsilon \geq 0.03$.

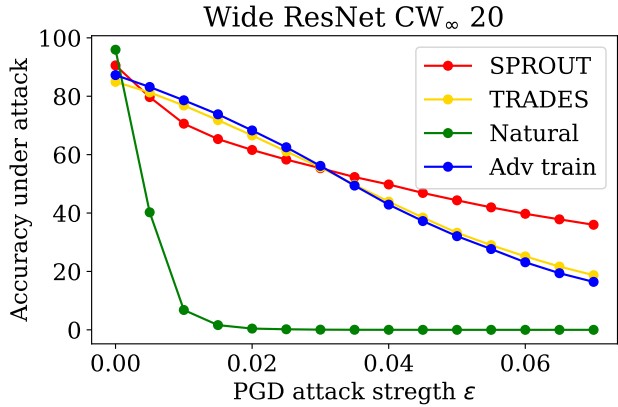

Figure 10: Robust accuracy under CW-$\ell_\infty$ attack.

### A.6    PERFORMANCE WITH DIFFERENT NUMBER OF RANDOM STARTS FOR PGD ATTACK

As suggested by (Madry et al., 2018), PGD attack with multiple random starts is a stronger attack method to evaluate robustness. Therefore, in Table 8, we conduct the following experiment on CIFAR-10 and wide ResNet to show that the model trained by SPROUT can still attain at least 61% accuracy against PGD-$\ell_\infty$ attack ($\epsilon = 0.03$) with the number of random starts varying from 1 to 10 and with 20 attack iterations. The robust accuracy of SPROUT is still clearly higher than other methods as shown in Figure 2. We also perform two additional attack settings: (i) 100-step PGD-$\ell_\infty$ attack with 10 random restarts using the CW loss and $\epsilon = 0.03$; (ii) 100-step PGD-$\ell_\infty$ attack with 10 random restarts using the cross entropy and $\epsilon = 0.03$. We find that SPROUT can still achieve 51.23% robust accuracy in setting (i) and 61.18% robust accuracy in setting (ii).

Table 8: Robust accuracy of SPROUT on PGD-$\ell_\infty$ attack with $\epsilon = 0.03$ using different number of random starts.

| # random start | 1 | 3 | 5 | 8 | 10 |
|---|---|---|---|---|---|
| Robust accuracy | 64.58% | 62.53% | 61.98% | 61.38% | 61.00% |

### A.7    PERFORMANCE COMPARISON WITH FREE ADVERSARIAL TRAINING ON RESNET-50 AND IMAGENET

Here we compare the performance of SPROUT with a pre-trained robust ResNet-50 model on ImageNet, which is shared by the authors in (Shafahi et al., 2019) proposing the free adversarial training method (Free Adv train). We find that SPROUT obtains similar robust accuracy as Free Adv train when $\epsilon \leq 0.01$. As $\epsilon$ becomes larger, Free Adv train has better robust accuracy.

Table 9: Robust accuracy under PGD-$\ell_\infty$ random targeted attack on ImageNet and ResNet-50

| Method | Clean Acc | $\epsilon = 0.005$ | $\epsilon = 0.01$ | $\epsilon = 0.015$ | $\epsilon = 0.02$ |
|---|---|---|---|---|---|
| Natural | 76.15% | 24.37% | 3.54% | 0.90% | 0.40% |
| Free Adv train | 60.49% | 51.35% | 42.29% | 32.96% | 24.45% |
| SPROUT | 61.23% | 51.69% | 38.14% | 25.98% | 18.52% |

