# OpenReview forum: "SPROUT: Self-Progressing Robust Training"
_ICLR.cc/2020/Conference — Reject_

### Official Review · AnonReviewer1 · 2019-10-22
**Official Blind Review #1**

**Rating:** 3

**Review:**

The authors proposed a hybrid method for defending against adversarial attacks called SPROUT. The proposed defense method consists of three main ingredients:

1. label smoothing with a learnable Dirichlet distribution
2. adding Gaussian noise to input examples
3. mixup: augment training examples with their linear combinations

The authors' main argument for their method is the speed over adversarial training and its effectiveness.

Individually, none of these ingredients are known to be strong defense against adversarial examples in the literature. Indeed this is corroborated by Figure 5, when the individual defenses do not have more than 10% accuracy under PGD100 attacks for epsilon=0.4. However, when all three are used together the accuracy jumps to close to 60%. This is very surprising. Another surprising fact is that in Figure 2, the method beats the benchmark adversarial PGD by more than 20% on white-box attacks, given the difficulty of beating adversarial PGD.

Given the surprise in these experimental results, I believe the authors should perform a more detailed analysis on how these ingredients for their SPROUT defense interact to produce such a strong predictor, in addition to doing ablation studies. An attempt should be made to explain why they work so well together when they are quite weak individually as defenses. It is difficult for me to recommend acceptance of this paper without an attempt to explain why it works.




**Experience Assessment:**

I have published one or two papers in this area.

**Review Assessment: Checking Correctness Of Derivations And Theory:**

I assessed the sensibility of the derivations and theory.

**Review Assessment: Checking Correctness Of Experiments:**

I assessed the sensibility of the experiments.

**Review Assessment: Thoroughness In Paper Reading:**

I read the paper thoroughly.

---

> ### Author Response · Authors · 2019-11-13
> **Response to Reviewer #1**
>
> We thank the reviewer for the precise summary of our work and for the keen observation on the complementary robustness of the three main ingredients. We totally agree with your comment that in addition to ablation studies, more detailed analysis is needed to explain their joint complementary robustness. In fact, in our original submission, we have already provided an explanation using the similarity analysis of the input gradients. Specifically, inspired by the diversity evaluation metric used in Kariyappa et al. for evaluating adversarial robustness of ensembles, in Appendix A.3 we have reported the pairwise cosine similarity of input gradients among the three ingredients in SPROUT (Dirichlet Label Smoothing, Gaussian Augmentation, and Mixup). We find that the cosine similarity between module pairs is indeed quite small (< 0.103), suggesting large diversity of these modules. We believe that this provides a strong implication: the diversified modules can provide complementary benefits to robustness improvement using our proposed co-training approach. In addition to diversity analysis, their complementary robustness can also be explained from each integradient’s unique contribution to model training. That is, Gaussian augmentation only perturbs the data samples, Dirichlet label smoothing only adjusts the training labels, and Mixup improves the generalization of the interpolated data samples based on the training data.
>
> We hope our responses addressed the reviewer’s concerns. We also would like to make the most of the openreview platform and are happy to take any additional questions the reviewer may have during the author rebuttal phase.

---

### Official Review · AnonReviewer2 · 2019-10-22
**Official Blind Review #2**

**Rating:** 6

**Review:**

This paper proposes a novel training method to build robust models. A new framework SPROUT is introduced to adjust label distribution during training. It also integrates mixup and Gaussian augmentation to further improve the robustness. The proposed method is built upon the Vicinity Risk Minimization (VRM) framework. Experiments show that the proposed method significantly outperforms the existing best methods in terms of robustness against attacks.

Overall, this paper proposes a novel method with good robustness performance. The proposed approach is built upon the VRM framework, and summarizes a lot of existing methods under this framework (Table 1). Experimental results are also very strong to prove the effectiveness of the proposed method.

On the other hand, I have some concerns about this paper. Since the performance improvement is significantly large over the current best methods, I need to see those concerns addressed to give a final rating.

1. How do you perform inference given testing data? Do you use Gaussian augmentation or mixup during inference?

2. Do you check that whether the robustness comes from obfuscated gradients? It's very important to examine the true robustness of the propose method.

3. What's the final distribution of \beta? Does it have a semantic meaning?

**Experience Assessment:**

I have published one or two papers in this area.

**Review Assessment: Checking Correctness Of Derivations And Theory:**

I carefully checked the derivations and theory.

**Review Assessment: Checking Correctness Of Experiments:**

I carefully checked the experiments.

**Review Assessment: Thoroughness In Paper Reading:**

I read the paper thoroughly.

---

> ### Author Response · Authors · 2019-11-13
> **Response to Reviewer #2**
>
> Response to AnonReviewer2:
> We thank the reviewer for acknowledging the contributions of our work. Please find our point-by-point response as follows:
>
> 1.We just use normal testing data in the inference time. We don’t make any changes on the testing data.
>
> 2.Yes. We have conducted several experiments to examize obfuscated gradients. Specifically, following the methods suggested in the ICML’18 paper “Obfuscated Gradients Give a False Sense of Security: Circumventing Defenses to Adversarial Examples”, we (1) vary the PGD attack iterations (Figure 6(a)); (2) report the robust accuracy with respect to different perturbation budgets (epsilon values); and (3) implement transfer attack to test our model (Table 2). Our robustness gain is comprehensive and consistently better than other methods.  In addition, in Figure 4 of Section 4.2, we have provided a visualization plot of the loss landscape with respect to the adversarial gradient direction and random direction. Among the hyperlane build by those two directions, our model achieves a much less loss compared with both adversarial training and TRADES. These results suggest that our robust training method does not cause obfuscated gradients.
>
> 3.Due to space limitation, in the original submission we have already provided some analysis about the learned label correlation from beta in Appendix A.2. In short, on CIFAR-10 we observed some clustering effect of class labels that are semantically close, and we also found the learned beta values are indeed not uniformly distributed.
>
> We hope our responses addressed the reviewer’s concerns. We also would like to make the most of the openreview platform and are happy to take any additional questions the reviewer may have during the author rebuttal phase.

---

### Official Review · AnonReviewer5 · 2019-11-01
**Official Blind Review #5**

**Rating:** 3

**Review:**

This work proposes training robust models without explicitly training on adversarial examples and by "smoothing" the labels in an adversarial fashion and by using Dirichlet label smoothing. Training robust models without adversarial training is indeed an important problem as mentioned by the authors since it can potentially (as the authors demonstrate) result in faster model training and less drop in clean accuracy. Overall the idea is interesting but I have some concerns mainly about evaluations and baselines which I am including below. If the authors can address my concerns, I am willing to increase my score:

1. Based on equations (9) and (10), if we set \alpha to be large, then the network is not trainable (since the worst-case adversary will increase the loss on the image by flipping the label during training). As a consequence, we can see that the value of hyper-parameters that the authors use is indeed very small (0.01 and 0.1). Even between these small values, the smaller value results in a better model. In the extreme case, where \alpha is zero there is no regularization and \beta becomes irrelevant. This illustrates that the performance of the model is very sensitive to \alpha. On the other hand, we can prevent the model from not learning anything by constraining \beta in equation (10) — similar to adversarial training where we constrain \delta. It seems that without constraining \beta, if the step-size for \beta is large, \beta can grow and completely mess up the labels even when \alpha is tiny. If we constrain \beta on the other hand, we can make sure that in no case, the top label for any augmented image is an incorrect label. Can the authors elaborate on why they did not set any limit on the value of \beta?
2. What is the batch-size used for ImageNet? The reason that I am asking is that you compute the gradient of \beta for the previous mini-batch but use it for the next mini-batch. Is it possible that the previous mini-batch's \beta is not accurate for the current mini-batch? For CIFAR, since the number of classes is 10, I would assume that you can update the statistics for the class (\betas) using the previous mini-batch since you always see examples from all the classes using any reasonable batch-size. What happens if you do the same but for a dataset with more classes but have the mini-batch be smaller than the number of classes. In this case, your \beta is getting updated only using information from a few classes and not all classes at once. In that case, what happens if you just use a random \beta every step?
3. For the white-box attacks, I also have a few questions. Do you use multiple random restarts? It is known that random restarts can be more effective than increasing the number of PGD attacks. See for example the leaderboards for MNIST and CIFAR-10 challenges by Madry. I would like to see a table where you plot how the accuracy changes by doing 100 step PGD attacks and by increasing the number of random restarts from 1 to 10 for example.
4. Do you do L-infinity CW attacks? I see that you have done L-2 CW attacks but I can't find any L-infinity CW attacks. It would be great to show numbers for that and also compare it with TRADES and PGD adversarial training. In previous smoothing methods, the L-infinity CW attack seems to be a stronger attack compared to PGD.
5. For the ImageNet task, the authors state that the evaluation of non-targeted attacks can result in label leaking. Label-leaking happens when one trains on adversarial examples built using a single-step attack and it means that the accuracy of adversarial examples is higher than natural examples at test-time. For this, I do not understand why the authors mention that they only evaluate targeted attacks while they are not doing any adversarial training.
6. Also, for ImageNet, there are recent methods such as Adversarial training for Free! where the authors do adversarial training on ImageNet with no overhead cost compared to natural training. Maybe this could be added as a better base-line than a naturally trained model.
7. In Figure 4. (a), why is the loss for the validation image illustrated so high? What image is this from the validation set?
8. In terms of Scalability, its good to mention new scalable methods such as YOPO and Adversarial Training for Free.
9. In the ablation study, including Dirichlet Smoothing indeed results in a huge boost compared to having no smoothing. However, it would be better to show that Dirichlet smoothing is indeed better than label-smoothing or adversarial smoothing by including results for other smoothing methods in Fig. 5.

**Experience Assessment:**

I have published in this field for several years.

**Review Assessment: Checking Correctness Of Derivations And Theory:**

I assessed the sensibility of the derivations and theory.

**Review Assessment: Checking Correctness Of Experiments:**

I carefully checked the experiments.

**Review Assessment: Thoroughness In Paper Reading:**

I read the paper thoroughly.

---

> ### Author Response · Authors · 2019-11-13
> **Response to Reviewer #5 (1/2)**
>
> We thank the reviewer for providing the review comments and suggestions. In the short rebuttal period, we have managed to include all the additional experiments suggested by the reviewer and updated the results in the revised version. Please find our point-by-point response as follows:
>
>
> 1. In SPROUT, \beta associates with the parameter of the Dirichlet distribution, which controls the statistical properties of generated label distributions. Specifically, consider the case z=Dirichlet(\beta). As described in equation (7), the mean of the s-th generated label value in z is proportional to the s-th entry of \beta divided by the total sum of the \beta entries. In other words, z=Dirichlet(\beta) generates a label distribution on the probability simplex, and the mean of z is \beta normalized by the sum of \beta entries. Therefore, in SPROUT we do not need to constrain the value of \beta, as the mean of the Dirichlet distribution will be properly normalized. Moreover, due to the normalization effect of the Dirichlet distribution, putting an additional constraint on \beta can be made equivalent to a particular \alpha value while keeping \beta unconstrained.
>
> 2. The batch size for ImageNet is 256. As described in Algorithm 1, when updating \beta we used the conventional stochastic optimization approach with the batch gradient. While it is possible that some classes are not sampled in a batch, similar to learning the model weights \theta, in the long run \beta can still be optimized properly based on stochastic optimization. Regarding the reviewer’s suggestion of using random \beta values, it is unclear to us what random functions should be used for a fair and meaningful comparison, given that random \beta values are not aiming to maximize the training loss during the iterations of model weight optimization process. Nonetheless, in the ablation study (Figure 5), we have shown that Dirichlet label smoothing (i.e., stochastic gradient ascent on \beta) significantly outperformed uniform label smoothing (i.e., fixed and uniform \beta values) in robust accuracy, which signifies the importance and effectiveness of stochastic optimization on \beta.
>
> 3. Following the reviewer’s suggestion, we have included Appendix A.6 in the revised version, where we set the number of random start from 1 to 10 and report the robust accuracy. Although there are some small performance variations, SPROUT can still achieve over 61% robust accuracy under PGD-Linfinity attack with epsilon=0.03 constraint, which clearly outperforms other methods.
>
> 4. Following the reviewer’s suggestion, we have included the results of CW-Linfinity attack in Appendix A.5 of the revised version. We find that the trend of robust accuracy is similar to that of PGD-Linfinity attack, where SPROUT shows a significant gain in robust accuracy for large epsilon values.
>
> 5. We agree with the reviewer that label leaking is not the right motivation in our setup, and we are sorry for the confusion. As many ImageNet class labels carry similar semantic meanings (e.g., different dog specifies as class labels), on ImageNet we follow the same setup as the ICML’18 paper “Obfuscated Gradients Give a False Sense of Security: Circumventing Defenses to Adversarial Examples” to generate meaningful adversarial examples for robustness evaluation using PGD-$\ell_\infty$ attacks with randomly targeted labels. We have revised the descriptions in our paper accordingly.

---

> ### Author Response · Authors · 2019-11-13
> **Response to Reviewer #5 (2/2)**
>
>
> 6. We thank the reviewer for bringing the paper “Adversarial Training for Free!” (Free Adv train) to our attention, which is a recently accepted paper to NeurIPS’19. We agree that it can be used as a good baseline for performance comparison, as it features similar robust accuracy to adversarial training with greatly reduced training time. In the revised version, we have included two sets of experiments as follows. (1) On CIFAR-10, we train the robust wide resnet 28 models using the default settings in the authors’ github. The performance comparison is added to Figure 2 and Table 5 of the revised version. In terms of robust accuracy, Free Adv train indeed has a similar performance as adversarial training. We also note that since Table 5 reports the 10-epoch run-time of each method, the advantage of Free Adv train over adversarial training may not be apparent, which we have emphasized in Section 4.5. (2) On ImageNet, we used the pre-trained robust ResNet-50 model shared by the authors and compared its robust accuracy in Appendix A.7 of the revised version.
>
> 7. We corrected an image plotting bug for producing Figure 4 (a) and have updated it in the revised version. The image used for loss landscape visualization is data sample #2233 in the testset.
>
> 8. Following the reviewer’s suggestion, in the revised version we have included the run-time analysis of “Adversarial Training for Free!” in Table 5 and Section 4.5.
>
> 9. Following the reviewer’s suggestion, in the revised version we have included the performance of uniform label smoothing in Figure 5.
>
>
> We hope our responses addressed the reviewer’s  concerns. We also would like to make the most of the openreview platform and are happy to take any additional questions the reviewer may have during the author rebuttal phase.

---

> > ### Comment · AnonReviewer5 · 2019-11-14
> > **Thanks and some more questions**
> >
> > Thank you very much for adding these extra experiments -- they were a lot and I really appreciate your responsiveness.  I hope that you also find that these new experiments could be helpful for evaluating the method which seems to work okay (doesn't seem to be more robust than TRADES and ADV training on CIFAR when attacking the CW loss but is extremely faster than them) for CIFAR-10 but not the best for ImageNet.
> >
> > After carefully reading your rebuttal and reviewing the revision, I have some other extra questions/comments. I understand that the rebuttal time is limited so I am going to prioritize them and have the more important ones at the top:
> >
> > a) According to A.6, the random restart experiments are done using a 20 step PGD attack on the cross-entropy loss. Can you please do 2 more extra experiments evaluating the robustness against 8/255 l-infinity attacks by doing 100-step PGD with 10 random restarts on the CW loss and also 100 step PGD on with 10 random restarts on the cross-entropy? It seems like the CW loss is a better objective for attacking the proposed smoothing method.
> >
> > b) According to A.7, the adversarially trained ImageNet is generally more robust. Do you have any thoughts on why this might be the case? On CIFAR-10 the results are very good but from A.7 the method doesn't seem to generalize to ImageNet with 1000 classes. Might this be because of the number of classes?
> >
> > c) Are the adversarially trained and TRADES models trained to resist eps=8/255? If yes, according to A.5, they are better for that eps (and also smaller perturbations).
> >
> > d) For Fig.5,  I meant adding label smoothing with all the other elements of SPROUT without Dirichlet (I apologize for the ambiguity -- this won't really affect my review given the limited rebuttal time left.). However, I do think it has value and it would be great if it is added to the future versions.

---

> > > ### Author Response · Authors · 2019-11-15
> > > **Thanks for the constructive questions and here are our responses.**
> > >
> > > Response to extra questions/comments:
> > >
> > > We also thank the reviewer for your responsiveness and efforts for reviewing our submission. We did find your comments very helpful in further strengthening our research findings and in improving the presentation of this paper. We have managed to perform all the extra comments the reviewer suggested. The point-to-point response is as follows:
> > >
> > > a) Following your suggestion, we have included two more experiments in Appendix A.6 of the revised version: (1)  100-step PGD-Linfinity attack with 10 random restarts on the CW loss and epsilon=0.03; (2) 100-step PGD-Linfinity attack with 10 random restarts on the cross-entropy and epsilon=0.03. We find that SPROUT could still achieve (1) 51.23% accuracy with 100-step PGD-Linfinity attack with 10 random restarts on the CW loss and (2) 61.18% accuracy with 10 random restart on cross-entropy loss. As the reviewer pointed out, since many papers have already identified the drop in robustness accuracy with more attack iterations or random starts, this trend will also be observed in other robust training methods such as Adv train and TRADES. For example, in the Madry’s Lab CIFAR-10 challenge leaderboard (https://github.com/MadryLab/cifar10_challenge), it is reported that 20-step PGD-Linfinity attack on the cross-entropy loss with 10 random restarts reduces the robust accuracy of Adv train to 45.21% (while SRPOUT attains 61%). In the interest of rebuttal deadline, we will report the full attack results of other methods in the next revision.
> > >
> > > b) There are several reasons that we believe can explain why the current ImageNet results of SPROUT are as not substantial as CIFAR-10 results. (1) First of all, due to limited rebuttal time and computation resource, we did not optimize the hyper-parameters (e.g. \alpha) of SPROUT for ResNet-50. Instead, we deploy the default settings of ResNet-152 (Table 3) for SPROUT training. We believe the robust accuracy of SPROUT can be improved with careful hyperparameter optimization. (2) Second, as the reviewer pointed out, the number of classes in ImageNet is indeed more than that of CIFAR-10. Nonetheless, in terms of training robust models, our results show that CIFAR-10 still has a large room for improvement. Moreover, many of the ImageNet class labels are semantically very similar (e.g., different dog specifies). Therefore, the \beta parameters along in Dirichlet distribution may not have sufficient expressive power to characterize the differences among semantically similar class labels in ImageNet. We expect that by incorporating more complex label smoothing functions, such as hierarchical Dirichlet distribution or Bayesian Dirichlet distribution, this question will be better understood. We also note that this extension still fits into the VRM framework of SPROUT and is a future direction that we will be exploring.
> > >
> > > c) Yes. We also note that the trend of robust accuracy is similar to Figure 2, where on ResNet SPROUT’s robust accuracy can be slightly worse than other methods when the epsilon value is small, while SPROUT becomes much more robust than others once the epsilon value passes a threshold. The observed threshold varies by network architectures and attack methods. For example, in the case of PGD-Linfinity attack on VGG the threshold is 0.01, and in the case of CW-Linfinity attack on ResNet the threshold is 0.03. We also note that the robust training of SPROUT is operated in a self-progressing manner, so different from adversarial training methods, we did not specify a perturbation threshold (nor an attack) to train a robust model.
> > >
> > > d) Following your suggestion, we have included the robust accuracy of uniform label smoothing+Gaussian augmentation+Mixup in FIgure 5 (with legend name GA+Mixup+LS). We find that SPROUT significantly outperforms GA+Mixup+LS (e.g. when epsilon = 0.03 our robust accuracy outperforms by at least 15%), implying the importance and effectiveness of Dirichlet label smoothing.

---

### Decision · Program_Chairs · 2019-12-19

**Decision:**

Reject

**Comment:**

This paper proposes a new training technique to produce a learned model robust against adversarial attacks -- without explicitly training on example attacked images. The core idea being that such a training scheme has the potential to reduce the cost in terms of training time for obtaining robustness, while also potentially increasing the clean performance. The method does so by proposing a version of label smoothing and doing two forms of data augmentations (gaussian noise and mixup).

The reviewers were mixed on this work. Two recommended weak reject while one recommended weak accept. All agreed that this work addressed an important problem and that the proposed solution was interesting. The authors and reviewers actively engaged in a discussion, in some cases with multiple back and forths. The main concern of the reviewers is the inconclusive experimental evidence. Though the authors did demonstrate strong performance on PGD attacks, the reviewers had concerns about some attack settings like epsilon and how that may unfairly disadvantage the baselines. In addition, the results on CW presented a different story than the results with PGD.

Therefore, we do not recommend this work for acceptance in its current form. The work offers strong preliminary evidence of a potential solution to provide robustness without direct adversarial training, but more analysis and explanation of when each component of their proposed solution should increase robustness is needed.